behaviour/ecology

Arctic mussel, *Mytilus* sp., shell growth, annual rhythm, valve behaviour, photoperiod

**Author for correspondence:**
Damien Tran
e-mail: damien.tran@u-bordeaux.fr

# Growth and behaviour of blue mussels, a re-emerging polar resident, follow a strong annual rhythm shaped by the extreme high Arctic light regime

Damien Tran[1,2], Hector Andrade[3], Guillaume Durier[1], Pierre Ciret[1,2], Peter Leopold[4,5], Mohamedou Sow[1], Carl Ballantine[3], Lionel Camus[3], Jørgen Berge[4,5,6] and Mickael Perrigault[1,2]

[1]University of Bordeaux, and [2]CNRS, EPOC, UMR 5805, F-33120 Arcachon, France
[3]Akvaplan-niva AS, Fram – High North Centre for Climate and the Environment, Langnes, Postbox 6606, 9296 Tromsø, Norway
[4]Faculty of Biosciences, Fisheries and Economics, UiT The Arctic University of Norway, 9037 Tromsø, Norway
[5]University Centre in Svalbard, Pb 156, 9171 Longyearbyen, Norway
[6]Department of Biology, Norwegian University of Science and Technology, Centre for Autonomous Marine Operations and Systems, NTNU

DT, 0000-0002-7432-7765

Polar regions are currently warming at a rate above the global average. One issue of concern is the consequences on biodiversity in relation to the Northward latitudinal shift in distribution of temperate species. In the present study, lasting almost two years, we examined two phenological traits, i.e. the shell growth and behavioural rhythm of a recently re-established species in the high Arctic, the blue mussel *Mytilus* sp. We compared this with a native species, the Islandic scallop *Chlamys islandica*. We show marked differences in the examined traits between the two species. In *Mytilus* sp., a clear annual pattern of shell growth strongly correlated to the valve behaviour rhythmicity, whereas *C. islandica* exhibited a shell growth pattern with a total absence of annual rhythmicity of behaviour. The shell growth was highly correlated to the photoperiod for the mussels but weaker for the scallops. The water temperature cycle was a very weak parameter to anticipate the phenology traits of both species. This study shows that the new resident in

the high Arctic, *Mytilus* sp., is a highly adaptive species, and therefore a promising bioindicator to study the consequences of biodiversity changes due to global warming.

## 1. Introduction

Arctic ecosystems undergo drastic climatic variability in terms of light, temperature and food availability on several scales, from daily changes to seasonal and annual ones. However, the linkages to biological processes still remain elusive [1,2]. In particular, the impact of extreme photoperiods within the Arctic on recent invasive species is largely unknown. Thus, limiting our ability to predict future changes in biodiversity and species abundances. The northward expansion of organisms at high latitudes raises the question of their phenotypic adaptation and the plasticity of their physiological mechanisms to extreme photoperiods [3,4]. Marine bivalves are an essential component of the benthic community. Due to their long-lived sessile lifestyle, bivalves are good indicators of Arctic ecosystem processes and its fast transition due to exacerbated global warming at the poles [5,6]. Blue mussels, a re-emerged resident in the high Arctic, have a distribution range limited mainly by winter air temperatures and thus are good candidates to study the adaptations of species undergoing a poleward expansion, informing on the consequences of climate change upon biodiversity [7–9].

In the present work, we analysed the seasonal timing (i.e. the phenology) of two biological processes that are the behaviour and the shell growth of the blue mussel *Mytilus* sp., a species which recently resettled in the Svalbard archipelago after a 1000-year absence. It was shown that the blue mussel present in Svalbard is a hybrid species, mixing three closely related species: *M. edulis*, *M. galloprovincialis* and *M. trossulus*. These three species didn't arrive on separate occasions in Svalbard, instead they have spread northwards gradually in a manner that allows genetic mixing [10–12]. In the absence of mussel native species, we studied in parallel, another species of resident bivalve, which occupied the same ecological niche as *Mytilus* sp. The scallop *Chlamys islandica*, which has been present in Svalbard at least since 7800–8500 years before present, was used in order to compare its behaviour with the new resident species [13,14]. We recorded continuously during almost two years the valve activity behaviour and the shell growth of both species in Kongsfjorden (Spitsbergen, Svalbard, 78°56′ N), a high Arctic predominantly ice-free fjord.

In this study, we present the first analysis, at an annual scale, of the valve behaviour jointly to the shell growth pattern of the mussel in the high Arctic. Through chronobiological analysis, we investigated the relationship between valve behaviour, growth and the photoperiod and water temperature. Finally, we highlighted differences and shared phenological traits studied in both bivalve species (native versus resettled).

## 2. Materials and methods

### 2.1. Animals, study area and experimental conditions

This study was conducted on 15 mussels (56.3 ± 3.7 mm shell length, mean ± effect size (ES)) collected in Longyearbyen, Svalbard and on seven scallops (63.4 ± 3.5 mm shell length, mean ± ES) collected in Kongsfjorden, Svalbard. The study was performed over a period of 712 days, from 5 May 2016 to 17 April 2018, encompassing two full polar day and polar night periods (electronic supplementary material, file S1). Both species were positioned in the same ballasted cage (50 × 100 cm) located at a depth of 4 m below the lowest average tidal range under an old pier in Ny-Alesund in Kongsfjorden (Spitsbergen Island, Svalbard; 78°56′ N, 11°56′ E). Valve behaviour was recorded *in situ* using two high-frequency non-invasive (HFNI) valvometers [15]. Briefly, a pair of lightweight electrodes were glued on each half shell. Between the electrodes, an electromagnetic current was generated, which allowed measurement of the amount of valve opening every 1.6 s. The data were transmitted daily to the laboratory using an Internet network. In parallel, water temperature was measured every 10 s (ADT7420 sensor, Analog Devices) near the valvometer.

### 2.2. Data treatment and statistical analyses

Hourly and daily valve opening amplitude (VOA, %) were analysed individually and then averaged on both species, with 0% and 100% of VOA corresponding respectively to the shell being closed and fully open. Double-plotted actograms of hourly VOA were produced during the 712-day experiment on both,

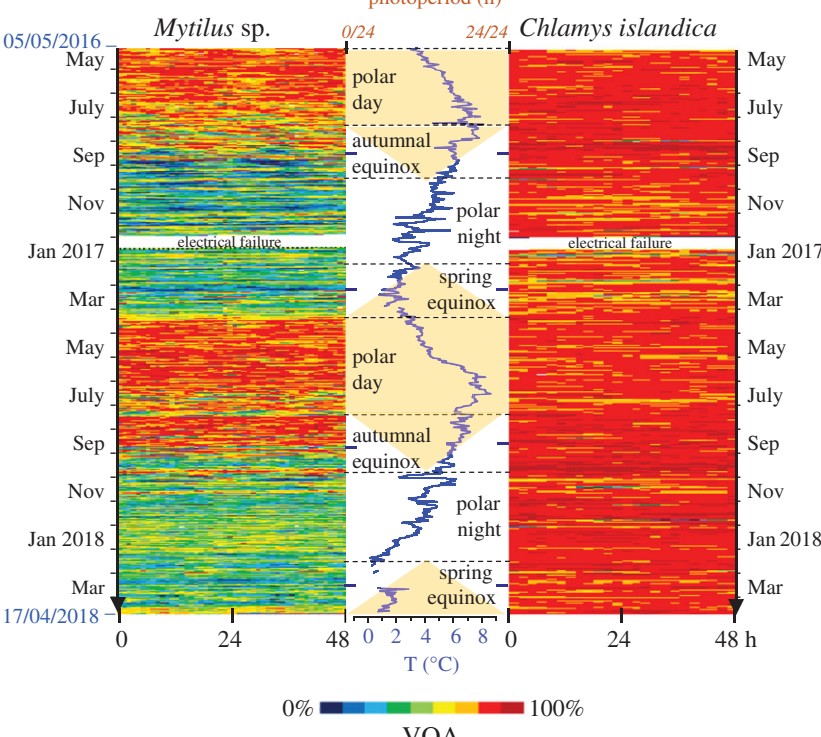

**Figure 1.** Annual pattern of valve opening amplitude (VOA). Actograms showed 712 days (from 5 May 2016 to 17 April 2018) of mean VOA of mussels (left panel, $n = 15$) and scallops (right panel, $n = 6$–$7$) in Kongsfjorden, Svalbard. Central panel, field temperature (blue line) and photoperiod (yellow: daylength, white: night length).

starting at 00.00 h (UTC). Each line represented 2 days (712 lines in our case). By convention, each 2-day period was first represented from 24 to 48 h, and then repeated on the next line from 0 to 24 h, and so forth. In bivalves, shell calcification takes place over the shell's internal surface in the mantle cavity. Consequently, the individual daily growth was calculated by measuring the minimal distance between electrodes when valves were closed. The relative slopes of shell growths were calculated in percentage according to the studied periods. Zero per cent and 100% of growth corresponded to the minimal distance in mm between the electrodes at the beginning and the end of the 712 days of the experiment, respectively.

Chronobiological analyses were performed with R using RAIN package [16] to investigate circannual periodicities of daily VOA and shell growth. The RAIN algorithm is a robust non-parametric method for the detection of rhythms, initially devoted to gene expression and adapted here for growth and behaviour data. The analyses were done on two-week data points of VOA or shell growth mean values, corresponding to 51 data points used with RAIN for each time series. Circannual periodicity was defined as a significant period of 364 days ± 14 days. To account for multiple testing (two sets of VOA data and two sets of growth data), Benjamini–Hochberg adjusted $p$-values less than 0.05 were considered significant [16]. Linear regressions and correlation coefficients (Pearson test) were performed after checking assumptions (normality and equal variance; $p$-value less than 0.05 considered significant) and using Sigma Plot software (v. 13.0; Systat Software, USA).

## 3. Results

Figure 1 presents actograms of mean daily valve opening amplitude (VOA, %) of *Mytilus* sp. and *C. islandica* during the 712-day experiment (individual actograms for each species are shown in the electronic supplementary material, file S2). The photoperiod and the water temperature are indicated between the actograms. The results show contrasting valve behaviour patterns. For the scallops, the VOA was near its maximum and stable regardless of the seasons. Conversely, the mussels expressed a divergent pattern of VOA, with a maximum during the polar day lasting until the autumnal equinox. Then, VOA was reduced from the autumnal equinox, continuing through the polar night and until the

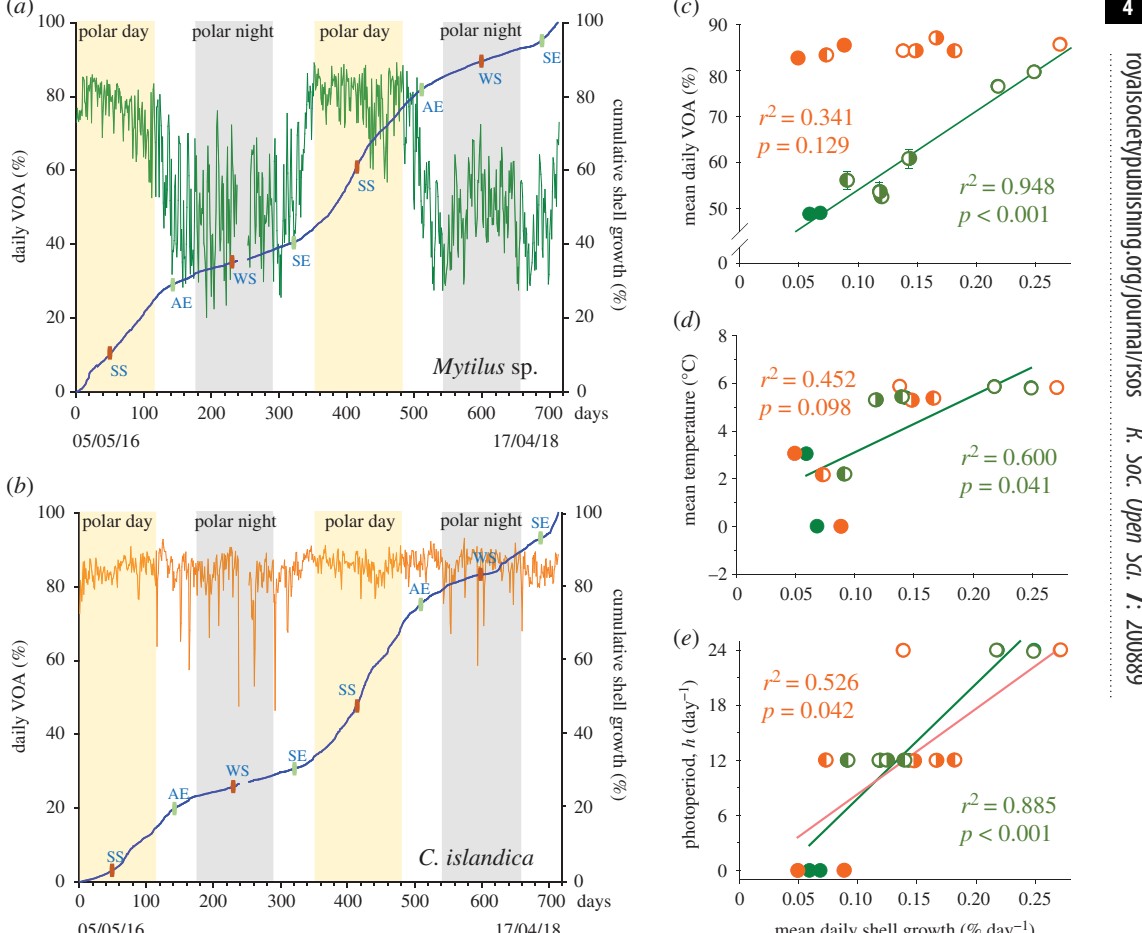

**Figure 2.** Shell growth and valve behaviour according to the photoperiod and temperature. Mean daily VOA (%) of (*a*) *Mytilus* sp. (green line) and (*b*) *C. islandica* (orange line) and mean daily cumulative shell growth (%, blue line) during the 712-day *in situ* experiment. SE: spring equinox; SS: summer solstice; AE: autumn equinox, WS: winter solstice. (*c,d,e*) Linear regression (*n* = 8) between the mean daily shell growth and the mean daily VOA, the mean water temperature and the photoperiod, respectively. In green symbols, mussels (*n* = 15); in orange symbols, scallops (*n* = 6). Linear regression shown for a significant *p*-value less than 0.05. Polar night: full circle; polar day: empty circle; light–dark alternation period around the spring: half full and half empty circle; light–dark alternation period around the autumn: half empty and half full circle.

beginning of the polar day. Figure 2 shows the daily mean VOA and the cumulative shell growth during the different seasons (individual profiles for both species of VOA and shell growth are shown in the electronic supplementary material, file S3). Again, two different patterns were observed. For *Mytilus* sp., the slope of the shell growth was correlated to VOA (figure 2*a*). Indeed, at the maximum VOA during the polar day, the growth slope was most pronounced. Contrarily, during the polar night, the VOA and the growth slopes were minimal. The transition of the growth slopes is centred on the spring and autumnal equinoxes, also corresponding to a transient VOA. The results for *C. islandica* (figure 2*b*) showed an absence of correlation between VOA and growth. The pattern of the growth slope is similar to *Mytilus* sp., with a maximum during the polar day and a minimum during the polar night, although the VOA remained remarkably stable. Linear regressions and correlation coefficients (Pearson tests, electronic supplementary material, file S4) between the shell growth and VOA (figure 2*c*), ambient temperature (figure 2*d*) and photoperiod (figure 2*e*) also revealed contrasting results between the species. For *Mytilus* sp., the shell growth was strongly correlated to VOA ($r^2 = 0.948$, $p < 0.001$) and photoperiod ($r^2 = 0.885$, $p < 0.001$), but less to temperature ($r^2 = 0.600$, $p = 0.041$). For *C. islandica*, the shell growth was not significantly correlated to VOA ($r^2 = 0.341$, $p = 0.129$) nor temperature ($r^2 = 0.452$, $p = 0.098$), but a low correlation was found with photoperiod ($r^2 = 0.526$, $p = 0.042$). The RAIN algorithm revealed a significant annual rhythmicity of VOA ($p = 3.5 \times 10^{-9}$) and shell growth ($p = 1.0 \times 10^{-16}$) for *Mytilus* sp. No circannual rhythm was detected for VOA in *C. islandica* ($p = 0.117$), but shell growth was highly significant ($p = 4.8 \times 10^{-9}$).

# 4. Discussion

Our findings clearly show contrasting results between the native scallop *C. islandica* and the re-emerging mussel *Mytilus* sp. in the high Arctic. First, regarding the pace for the curve of the shell growth, both species exhibited a similar annual pattern. Second, we showed a clear annual rhythm of valve behaviour highly correlated to the annual rhythm of the shell growth in the mussels, but not in the scallops. Finally, we demonstrated that shell growth was strongly correlated to the photoperiod for the mussels, but less so for the scallops.

In *Mytilus* sp., we observed a clear annual pattern of shell growth which follows the annual drastic photoperiod cycle, for which the high Arctic is known, correlating well to the behavioural rhythm. By comparison, in the Artic bivalve *Arctica islandica*, it was suggested that photoperiod is a cue of the annual valve behaviour, yet it is weaker than that of food supply [17]. This annual rhythm is shared among bivalves living in the oceanic temperate zone [18]. Indeed, photoperiod is known as the primary environmental cue that drives annual rhythms and is also proposed to be the zeitgeber of a putative circannual clock responsible for generating endogenous rhythmicity at the annual scale [7,8]. However, as previously shown [13], *C. islandica* didn't follow an annual rhythm of valve behaviour, although an annual rhythm of shell growth was exhibited. We suspect that the mantle edge of the scallop, which delimits the real aperture of the shell, follows an annual rhythm. In this case, the mantle edge will be regulating the water flow rate passing through the pallial cavity, adapting to the food availability during the year. Indeed, different mechanisms are responsible for controlling gill filtration in both species, which might explain the correlations between valve behaviour and annual rhythm. In blue mussels, the valve aperture controls the amount of water filtrated per unit of time, and so the capacity to ingest nutritive particles. In other words, VOA controls water flow rate. For scallops in contrast, the valves remain open and it is the very flexible mantle edge that is responsible for controlling the water flow rate, as mentioned above. In terms of muscles, the abductor muscles control shell closures in bivalves through smooth and striated muscle bundles. For mussels, the smooth muscle is well developed, allowing the shells to remain closed during long time periods without a major energy cost. For scallops it is the contrary; the smooth muscle is weak which hinders long-term shell closures. Therefore, remaining with the valves open and regulating the aperture with the mantle edge is more effective in terms of energy efficiency. Lastly, with regard to environmental cues, water temperature seemed a poor proxy of phenological traits in both species, with a low or absent correlation to shell growth in *Mytilus* sp. and *C. islandica*, respectively.

We showed that the mussel growth rate along with VOA was maximal between the spring and autumnal equinoxes, with a total maximum around the summer solstice during the polar day. Inversely, the growth rate decreases from the autumnal equinox to the spring equinox, with the lowest slope at the winter solstice during the polar night. However, we have shown that significant shell growth occurs during the darkest period of the polar night, even if it is approximately four times slower than during maximal growth throughout the polar day. This result reinforced the new paradigm proposed, stating that during the polar night, biological activity levels still remain important [19]. In *C. islandica*, the curve of the annual growth is less correlated to the annual photoperiod (figure 2*e*). Two different strategies might have been adopted. *Mytilus* sp. might have reduced its metabolism during the adverse conditions of polar night, when food availability is limited. Experimental starvation studies have shown that *Mytilus edulis* reduces valve opening to reduce metabolism as a means of overcoming starvation periods while at the same time increasing shell length [20,21]. In our results, shell growth continued during the polar night, albeit at a lower rate. This is in concordance with previous studies in Greenland showing a reduction of shell growth during the polar night in the Arctic bivalves *Serripes groenlandicus* and *Clinocardium ciliatum* [22]. On the contrary, *C. islandica* kept a more active behaviour during the polar night. This strategy seems shared by the Arctic amphipod *Onisimus litoralis*, which is growing at the same rate throughout the year [23]. *Chlamys islandica* may be able to feed on more diversified and smaller particulate food resources than *Mytilus* sp. This scallop may be able to use an autotrophic food resource such as bacteria which is available even during the polar night [24], which the blue mussel would not be able to do.

In conclusion, *Mytilus* sp. appears as a very tolerant and adaptive species, probably at the onset of a new resettlement phase in high Arctic. The effects of extreme photoperiods on the annual rhythm of this species are a negligible factor on the northward expansion, which in turn reflects this species' high capacity to deal with drastic light regimes. Due to climate warming, species like *Mytilus* sp. are able to shift their distribution northwards into the high Arctic. In the near future, the issue that will be raised is the impact of re-emerging *Mytilus* sp. on local species that share the same ecological niche.

Ethics. All experiments complied with the laws in effect in Svalbard and they conformed to international ethical standards.

Data accessibility. The data underlying this study are available in the electronic supplementary material (SM_Row data_dailyVOA&growth_Tran *et al._*2020).

Authors' contributions. D.T., M.P., P.C., L.C., J.B., P.L. and H.A. were involved in study design; D.T., M.P., P.C., C.B., P.L. and H.A. were involved in fieldwork; D.T., G.D. and M.S. were involved in data treatment; D.T. and G.D. were involved in interpretation; D.T. was involved in manuscript writing; D.T., L.C. and H.A. were involved in funding. All authors contributed critically to the drafts, and gave final approval for publication.

Competing interests. We have no competing interests.

Funding. The funds were provided by the French National Research Agency (WAQMOS project 15-CE04-0002), the French Polar Institute (ARCTICLOCK project 1166), the Svalbard Environmental Protection Fund (project 15/133) and the High North Research Centre for Climate and the Environment (Fram Centre) throughout the flagship 'Effects of climate change on sea and coastal ecology in the north'.

Acknowledgements. The authors thank P.E. Renaud for his assistance in collecting scallops and L. Payton for constructive discussion.

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
