## [Reviewer comments · Royal Society Open Science]

Review History

RSOS-200889.R0 (Original submission)

Review form: Reviewer 1

Is the manuscript scientifically sound in its present form?

No

Are the interpretations and conclusions justified by the results?

No

Is the language acceptable?

No

Do you have any ethical concerns with this paper?

No

Have you any concerns about statistical analyses in this paper?

Yes

Recommendation?

Major revision is needed (please make suggestions in comments)

Comments to the Author(s)

The study aims to explain how species immigrating to polar latitude may be challenged by its extreme environment. This is done by comparing blue mussels that have recently established in Svalbard to Icelandic scallops. This paper presented interesting data on the annual behavior and growth of the two bivalves at high latitude. The data provided evidence for divergent strategies used by very different species of bivalves above the polar circle across the annual cycle, which furthers our understanding of how different species cope with the extreme environmental conditions near the poles.

I can very much appreciate the difficulty and patience needed to collect data over such an extended duration. There are, however, a few deficiencies that prohibited me from evaluating the manuscript appropriately and that indicate the manuscript is based on a weak conceptual foundation.

--Language

Multiple language mistakes made it difficult to grasp the full intent of the authors and were distracting. Such interesting data would benefit from a coherent presentation, so that readers can fully appreciate their significance. A few limited examples are:

Lines 93-94

Lines 134-136

Line 145

The supplementary material also has language mistakes.

--Conceptual foundation

The conceptual foundation needs improvement, and the paper would benefit from a substantial restructuring. The Arctic is a climatic zone while polar regions are defined by photoperiod, and blue mussels have a long history of being present above the polar circle where continuous light and dark are present. Also, the authors never identify the species of blue mussel in the study, and they only refer to the mussel as a *Mytilus* sp., so we do not know whether these individuals are from source populations from below the polar circle. This information seems necessary to address adequately the motivation of the study that blue mussels are a good indicator of how species newly immigrating to polar latitude cope with extreme photoperiod. This also affects the interpretation of the data in the discussion and conclusions.

Additionally, readers may be a little confused with the comparison of the scallop with the mussel without a description of their shared and individual evolutionary history. How long has the scallop resided in the high Arctic? Is this meaningfully different from *Mytilus* spp.? How closely are the two species related? Relevant information about the scallop should be included in the introduction.

Some may find it more surprising that the scallops do not follow the annual cycle. It is unfortunate that experimental limitations seem to preclude making any conclusions about attenuation of their cyclic behavior.

--Statistics

The double plotted actograms are very useful in helping the reader understand the data, and my main concerns regard the linear models.

How do the authors rectify the probable correlation between photoperiod and temperature? These two environmental parameters are highly correlated in polar regions. The result should be discussed in this context, with particular emphasis on the collinearity of temperature and photoperiod and the seasonal influence that the photic environment has on temperature. The weaker signal from temperature for both the scallop and blue mussel seems related to the

increased variability in temperature compared to photoperiod. Making conclusions based on photoperiod alone seems misleading, because photoperiod and temperature are positively correlated.

How was temporal autocorrelation accounted for in the linear models?

Were temperature and photoperiod tested in separate linear models, or were they both included in a single model?

The data in figure 2 D are suggestive of a non-linear relationship, and a polynomial predictor should be used to investigate the relationship for both species. This can be done using R.

Line 134 states that results for temperature and photoperiod were contrasting - contrasting what? The results complement each other for blue mussels; they both indicate positive correlations with mean daily shell growth (Fig. 2D,E). For the scallops, the results seem similar to blue mussels (Fig. 2E), especially if the relationship is non-linear.

Please report R-square values and effect sizes for all linear models because p values do not provide information about the biological significance of a result.

Review form: Reviewer 2

Is the manuscript scientifically sound in its present form?

Yes

Are the interpretations and conclusions justified by the results?

No

Is the language acceptable?

Yes

Do you have any ethical concerns with this paper?

No

Have you any concerns about statistical analyses in this paper?

Yes

Recommendation?

Accept with minor revision (please list in comments)

Comments to the Author(s)

Review Tran et al.

This is a nice piece of work with a strait forward experimental set up. The only point which I do not understand in this set up is why the authors chose the scallop *Chlamys islandica* as examples for a native species and not a native mussel with a similar valve activity behavior to *Mytilus* sp.. This would have increased the informative value to what extent *Mytilus* is able to establish itself in higher latitudes. The choise of the organisms used should be explained in the text.

Abstract

Row 37: ...in the high Arctic,...

Introduction

Row 52: ...in the Arctic...

Row 57...of Arctic ecosystem processes

Results

Figure 1: It would be informative when on the right and left side are the month of the year

Row 135:.....revealed contrasting results between the species.

Row 137: ...lesser to temperature.

Row 139: ...with photoperiod.

Discussion

Row 145:...show contrasting results..

Row 149:...we demonstrated. (to choose another word than always showed)

Row 151-156: This sentence is in combination with the following confusing. I would recommend to remove the first sentence and start with: In *Mytilus* sp., we observed a clear annual pattern of shell growth that follows.....In the Arctic bivalve *Arctica islandica*, it was suggested that photoperiod is a cue for the annual valve behavior but weaker than food supply. This annual valve rhythm is shared...

Conclusion

Row 202.204: "The effects of extreme photoperiods on the annual rhythm of this species is a negligible factor on the species northward expansion and reflects the species high capacity to deal with drastic light regimes"

To conclude that *Mytilus* sp. Has a high capacity to deal with the drastic light regime is in my opinion only possible when comparing the data with native mussels of a similar valve behavior pattern. Therefore, the authors has to be clear that this is nota direct comparision between the two species (see comment in the intro. above), so that they should consider this point in their conclusion.

Decision letter (RSOS-200889.R0)

Dear Dr Tran,

The editors assigned to your paper ("Growth and behavior of blue mussels, a reemerging polar resident, follow a strong annual rhythm shaped by the extreme high Arctic light regime") have now received comments from reviewers. We would like you to revise your paper in accordance with the referee and Associate Editor suggestions which can be found below (not including confidential reports to the Editor). Please note this decision does not guarantee eventual acceptance.

Please submit a copy of your revised paper before 15-Aug-2020. Please note that the revision deadline will expire at 00.00am on this date. If we do not hear from you within this time then it will be assumed that the paper has been withdrawn. In exceptional circumstances, extensions may be possible if agreed with the Editorial Office in advance. We do not allow multiple rounds of revision so we urge you to make every effort to fully address all of the comments at this stage. If deemed necessary by the Editors, your manuscript will be sent back to one or more of the original reviewers for assessment. If the original reviewers are not available, we may invite new reviewers.

- Data accessibility

If you wish to submit your supporting data or code to Dryad (<http://datadryad.org/>), or modify your current submission to dryad, please use the following link:
<http://datadryad.org/submit?journalID=RSOS&manu=RSOS-200889>

- Competing interests

- Authors' contributions

- Acknowledgements

- Funding statement

on behalf of Dr Jeffrey Thompson (Associate Editor) and Pete Smith (Subject Editor)
openscience@royalsociety.org

Associate Editor's comments (Dr Jeffrey Thompson):

Comments to the Author:

Hello,

Following review of this manuscript by two external reviewers, I recommend major revisions prior to resubmission. Both reviewers have made a number of comments which I think can help to contribute to a clearer and improved manuscript.

Reviewer #1 has raised a few issues that need to be addressed prior to review. If possible, they argue it would be worth identifying the species of *Mytilus* used, or at least its source population, which is important for the conceptual framing the manuscript. Additionally, Reviewer #1 recommends some modifications/additional discussion regarding the linear models used by the authors, and possible autocorrelation that may be present in the datasets. Additionally, and I agree with them on this point, they ask for R-squared and effect sizes for the linear models.

Reviewer #2 asks for details relating to the choice of organisms used for the text, as well as points out a number of stylistic changes that need to be made to the text.

I suggest that the authors look carefully at both review sheets from the reviewers in preparing a revised submission.

Reviewer #1 also points out a number of grammatical issues, which the authors should address prior to resubmission.

Reviewers' Comments to Author:

Reviewer: 1

Comments to the Author(s)

The study aims to explain how species immigrating to polar latitude may be challenged by its extreme environment. This is done by comparing blue mussels that have recently established in Svalbard to Icelandic scallops. This paper presented interesting data on the annual behavior and growth of the two bivalves at high latitude. The data provided evidence for divergent strategies used by very different species of bivalves above the polar circle across the annual cycle, which furthers our understanding of how different species cope with the extreme environmental conditions near the poles.

I can very much appreciate the difficulty and patience needed to collect data over such an extended duration. There are, however, a few deficiencies that prohibited me from evaluating the manuscript appropriately and that indicate the manuscript is based on a weak conceptual foundation.

--Language

Multiple language mistakes made it difficult to grasp the full intent of the authors and were distracting. Such interesting data would benefit from a coherent presentation, so that readers can fully appreciate their significance. A few limited examples are:

Lines 93-94

Lines 134-136

Line 145

The supplementary material also has language mistakes.

--Conceptual foundation

The conceptual foundation needs improvement, and the paper would benefit from a substantial restructuring. The Arctic is a climatic zone while polar regions are defined by photoperiod, and blue mussels have a long history of being present above the polar circle where continuous light and dark are present. Also, the authors never identify the species of blue mussel in the study, and they only refer to the mussel as a *Mytilus* sp., so we do not know whether these individuals are from source populations from below the polar circle. This information seems necessary to address adequately the motivation of the study that blue mussels are a good indicator of how species newly immigrating to polar latitude cope with extreme photoperiod. This also affects the interpretation of the data in the discussion and conclusions.

Additionally, readers may be a little confused with the comparison of the scallop with the mussel without a description of their shared and individual evolutionary history. How long has the scallop resided in the high Arctic? Is this meaningfully different from *Mytilus* spp.? How closely are the two species related? Relevant information about the scallop should be included in the introduction.

Some may find it more surprising that the scallops do not follow the annual cycle. It is unfortunate that experimental limitations seem to preclude making any conclusions about attenuation of their cyclic behavior.

--Statistics

The double plotted actograms are very useful in helping the reader understand the data, and my main concerns regard the linear models.

How do the authors rectify the probable correlation between photoperiod and temperature? These two environmental parameters are highly correlated in polar regions. The result should be discussed in this context, with particular emphasis on the collinearity of temperature and photoperiod and the seasonal influence that the photic environment has on temperature. The weaker signal from temperature for both the scallop and blue mussel seems related to the increased variability in temperature compared to photoperiod. Making conclusions based on photoperiod alone seems misleading, because photoperiod and temperature are positively correlated.

How was temporal autocorrelation accounted for in the linear models?

Were temperature and photoperiod tested in separate linear models, or were they both included in a single model?

The data in figure 2 D are suggestive of a non-linear relationship, and a polynomial predictor should be used to investigate the relationship for both species. This can be done using R.

Line 134 states that results for temperature and photoperiod were contrasting - contrasting what? The results complement each other for blue mussels; they both indicate positive correlations with mean daily shell growth (Fig. 2D,E). For the scallops, the results seem similar to blue mussels (Fig. 2E), especially if the relationship is non-linear.

Please report R-square values and effect sizes for all linear models because p values do not provide information about the biological significance of a result.

Reviewer: 2

Comments to the Author(s)
Review Tran et al.

This is a nice piece of work with a strait forward experimental set up. The only point which I do not understand in this set up is why the authors chose the scallop *Chlamys islandica* as examples for a native species and not a native mussel with a similar valve activity behavior to *Mytilus* sp.. This would have increased the informative value to what extent *Mytilus* is able to establish itself in higher latitudes. The choise of the organisms used should be explained in the text.

Abstract

Row 37: ...in the high Arctic,...

Introduction

Row 52: ...in the Arctic...

Row 57...of Arctic ecosystem processes

Results

Figure 1: It would be informative when on the right and left side are the month of the year

Row 135:.....revealed contrasting results between the species.

Row 137: ...lesser to temperature.

Row 139: ...with photoperiod.

Discussion

Row 145:...show contrasting results..

Row 149:...we demonstrated. (to choose another word than always showed)

Row 151-156: This sentence is in combination with the following confusing. I would recommend to remove the first sentence and start with: In *Mytilus* sp., we observed a clear annual pattern of shell growth that follows.....In the Arctic bivalve *Arctica islandica*, it was suggested that photoperiod is a cue for the annual valve behavior but weaker than food supply. This annual valve rhythm is shared...

Conclusion

Row 202.204: "The effects of extreme photoperiods on the annual rhythm of this species is a negligible factor on the species northward expansion and reflects the species high capacity to deal with drastic light regimes"

To conclude that *Mytilus* sp. Has a high capacity to deal with the drastic light regime is in my opinion only possible when comparing the data with native mussels of a similar valve behavior pattern. Therefore, the authors has to be clear that this is nota direct comparision between the two species (see comment in the intro. above), so that they should consider this point in their conclusion.

Author's Response to Decision Letter for (RSOS-200889.R0)

See Appendix A.

Decision letter (RSOS-200889.R1)

Dear Dr Tran,

It is a pleasure to accept your manuscript entitled "Growth and behavior of blue mussels, a reemerging polar resident, follow a strong annual rhythm shaped by the extreme high Arctic light regime" in its current form for publication in Royal Society Open Science.

Additionally, we note that the following email address is currently marked as invalid. Please send to the Editorial Office an updated email for the following:

- mickael.perrigault@u-bordeaux.fr

on behalf of Dr Jeffrey Thompson (Associate Editor) and Pete Smith (Subject Editor)
openscience@royalsociety.org

Appendix A

Ms. No. RSOS-200889

Title : « Growth and behavior of blue mussels, a reemerging polar resident, follow a strong annual rhythm shaped by the extreme high Arctic light regime »

Response to Associate Editor and Referees.

Associate Editor

Associate Editor's comments (Dr Jeffrey Thompson):

Comments to the Author:

Hello,

Following review of this manuscript by two external reviewers, I recommend major revisions prior to resubmission. Both reviewers have made a number of comments which I think can help to contribute to a clearer and improved manuscript.

Reviewer #1 has raised a few issues that need to be addressed prior to review. If possible, they argue it would be worth identifying the species of *Mytilus* used, or at least its source population, which is important for the conceptual framing the manuscript.

Additionally, Reviewer #1 recommends some modifications/additional discussion regarding the linear models used by the authors, and possible autocorrelation that may be present in the datasets.

Additionally, and I agree with them on this point, they ask for R-squared and effect sizes for the linear models.

Reviewer #2 asks for details relating to the choice of organisms used for the text, as well as points out a number of stylistic changes that need to be made to the text.

I suggest that the authors look carefully at both review sheets from the reviewers in preparing a revised submission.

Reviewer #1 also points out a number of grammatical issues, which the authors should address prior to resubmission.

We want to thank the associate editor for allowing us the opportunity to revise the manuscript and submit a new version.

Following the associate editor and the referee's advice, we have taken into account their recommendations and revised the manuscript. The modifications are indicated in blue font and referred to the line numbers in the revised version.

We hope that this new, revised version of the manuscript has addressed the editor and referee's concerns in a satisfactory manner, making the manuscript suitable for publication in the journal.

Best regards

The authors

Reviewers' Comments to Author:

Reviewer: 1

Comments to the Author(s)

The study aims to explain how species immigrating to polar latitude may be challenged by its extreme environment. This is done by comparing blue mussels that have recently established in Svalbard to Icelandic scallops. This paper presented interesting data on the annual behavior and growth of the two bivalves at high latitude. The data provided evidence for divergent strategies used by very different species of bivalves above the polar circle across the annual cycle, which furthers our understanding of how different species cope with the extreme environmental conditions near the poles.

I can very much appreciate the difficulty and patience needed to collect data over such an extended duration. There are, however, a few deficiencies that prohibited me from evaluating the manuscript appropriately and that indicate the manuscript is based on a weak conceptual foundation.

We thank the referee to her / his valuable comments. Concerning the language, we have (one co-author, Carl Ballantine, is an English native speaker) carefully read the manus and edited it to improve its readability. Concerning the statistics and the conceptual foundation, we have taken into account the advices of the referee and modified our manus to improve this.

--Language

Multiple language mistakes made it difficult to grasp the full intent of the authors and were distracting. Such interesting data would benefit from a coherent presentation, so that readers can fully appreciate their significance.

We have edited the manus. In addition to the examples highlighted by the referee we have modified several sentences:

Ln 27, 49, 51-53, 56-58, 76, 84, 87, 102, 107, 112-114, 123, 134-135, 135, 137-138, 154, 171, 179-182, 188-190, 202-205, 208-210, 234.

A few limited examples are:

Lines 93-94

Ok, we modified the sentence, ln 98-99

Lines 134-136

Ok, we modified the sentences, ln 134-138

Line 145

Ok, we modified the sentence, ln 143-147

The supplementary material also has language mistakes.

With accordance to the referee, we checked the supplementary material. We modified the legend of the table S1.

--Conceptual foundation

The conceptual foundation needs improvement, and the paper would benefit from a substantial restructuring. The Arctic is a climatic zone while polar regions are defined by photoperiod, and blue mussels have a long history of being present above the polar circle where continuous light and dark are present.

Also, the authors never identify the species of blue mussel in the study, and they only refer to the mussel as a *Mytilus* sp., so we do not know whether these individuals are from source populations from below the polar circle. This information seems necessary to address adequately the motivation of the study that blue mussels are a good indicator of how species newly immigrating to polar latitude cope with extreme photoperiod. This also affects the interpretation of the data in the discussion and conclusions.

Due to funding constraints, no genetic analyses were performed to identify to the species level for this study.

However, other studies indicate that in Svalbard we have hybrid blue mussels. Our sample set of mussels come from this population present at least since 2004.

The first observation of living blue mussels in Svalbard was made in 2004, after a last colonization one thousand years ago. Several hypotheses were given to explain the origin of the blue mussels in Svalbard. Genetic studies indicated that the species *M. edulis* had spread northwards from northern Norway through a natural invasion from the south, mainly due to the Gulf stream which warms and enters the fjords of the west coasts of the Svalbard. The warm water had also brought southern species *Mytilus galloprovincialis*, from the Mediterranean Sea. Finally, a third species is present, the North American species (*M. trossolus*), which was common mainly along the coast of Greenland. In fact, the analysis of genetic materials of the mussels in Svalbard have shown that we have mussel hybrids, a mixture of species (*M. edulis*, *M. galloprovincialis* and *M. trossolus*) and no single species. It is thought that the 3 species didn't arrive separately in Svalbard, but they have spread northwards gradually in a manner that allows genetic mixing. Only two species have headed northwards, the species found in the US and Greenland has somehow managed to expand eastwards towards Svalbard. That is unless it is also found in Northern Norway and has managed to hybridize there as well. The hypothesis of invasion via shipping is refuted, due to the genetic composition being similar between the separate populations found in different fjordic localities on Svalbard.

For more information about this issue, see the articles cited in the manus: Ref 11 &12.

Mathiesen SS, Thyrring J, Hemmer-Hansen J, Berge J, Sukhotin A, Leopold P, Bekaert M, Sejr MK, Nielsen EE. (2016) Genetic diversity and connectivity within Mytilus spp. in the subarctic and Arctic. Evol. Appl. 10: 39-55, doi: 10.1111/eva.12415

Leopold P, Renaud PE, Ambrose WG, Berge J. 2019 High Arctic Mytilus spp.: occurrence, distribution and history of dispersal. Polar Biol. 42, 237–244. (doi:10.1007/s00300-018-2415-1)

New sentences are added in 64-68.

Moreover, *Mytilus* is commonly used as climate indicators with winter air temperature being one of the main factors limiting their distribution and with several authors referring to the

species as a sensitive indicator for climate change (Leopold et al. 2019 and references therein).

The sentence is modified ln 58-61.

Additionally, readers may be a little confused with the comparison of the scallop with the mussel without a description of their shared and individual evolutionary history. How long has the scallop resided in the high Arctic? Is this meaningfully different from *Mytilus* spp.? How closely are the two species related? Relevant information about the scallop should be included in the introduction.

The specie *Chlamys islandica* is a resident in Svalbard, at least since 7800-8500 years BP (before present) (Rowan et al. 1982).

Both species are related due to their place in the food web structure. Both species are fixed by a byssus to a hard substrate and have mainly the same source of feeding. However, one species is a mussel and the other one is a scallop. The referee 1 had the same concern. It would have been better to compare two species of mussels, a native versus a new resident, but it was not possible. No other mussel species are present in Svalbard. And for example, the mussel *Modiolus modiolus* present lower in Arctic, is not resident currently in Svalbard.

We added information in the introduction and modified the sentences ln 68-71.

We added as well a new reference, now Ref. 14: Rowan et al. 1982. *Holocene Glacial Geology of the Svea Lowland, Spitsbergen, Svalbard. Geografiska Annaler. Series A, Physical Geography Vol. 64, No. 1/2 (1982), pp. 35-51.*

Some may find it more surprising that the scallops do not follow the annual cycle. It is unfortunate that experimental limitations seem to preclude making any conclusions about attenuation of their cyclic behavior.

The scallop follows an annual cycle but not in terms of valve activity. We explain in the Discussion section (ln 166-182) that instead of valve aperture, it would be the mantle edge, very retractile, that regulates the water flow rate and so could follow an annual rhythm. Long-term video recording would be necessary to validate this hypothesis. However, it remains very difficult to collect such data in situ nevermind over a prolonged one-year experiment.

--Statistics

The double plotted actograms are very useful in helping the reader understand the data, and my main concerns regard the linear models.

How do the authors rectify the probable correlation between photoperiod and temperature?

These two environmental parameters are highly correlated in polar regions.

The result should be discussed in this context, with particular emphasis on the collinearity of temperature and photoperiod and the seasonal influence that the photic environment has on temperature. The weaker signal from temperature for both the scallop and blue mussel seems related to the increased variability in temperature compared to photoperiod. Making conclusions based on photoperiod alone seems misleading, because photoperiod and temperature are positively correlated.

We agree that photoperiod and temperature are correlated, not only in polar regions. It is the sun light that is at the origin of temperature fluctuations, and yes the annual temperature pattern is more variable than the photoperiod. That is why it is the photoperiod and not temperature that synchronize the internal annual clock at the origin of the seasonal / annual behavior of the animals. Light is a better and much more robust environmental cue than temperature. Photoperiod is not directly dependent from the climate change on earth, being solely dependent on the earth and sun cycles.

The photoperiodism in the annual rhythm generation is largely documented and well-established, especially in temperate areas. We show that in polar regions, when an annual rhythm is present it is also well correlated to photoperiod for some species.

We know that the photoperiod is at the origin of the synchronization of the biological rhythm (causality dependence) and the temperature (causality dependence). But the existence of a relation between annual pattern of temperature and biological rhythm would be more a correlation than a direct causal interaction.

To take into account the remark of the referee, we did a test of correlation (Pearson coefficients) for each species. These tables will be added as supplemental information (file S4, see tables below). The results show of course a correlation between the photoperiod and the temperature ($r = 0.781$, $p = 0.0381$) but confirm the results showed Fig. 2 C-E.

Concerning the scallop, no correlations were found between the growth, VOA, and photoperiod or temperature. On contrary concerning the mussels, it was found a correlation between VOA and the growth or the photoperiod, and a high correlation between the growth and the photoperiod and a low correlation with temperature. These results highlighted again a high importance of photoperiod but a weaker or null effect of temperature to explain VOA or the growth.

We modified the sentences In 116-117 and 137-138.

Table S4. Tables of Pearson's correlation coefficients. A: the scallop *C. islandica*, B: the mussel *Mytilus sp.* Each cell contents the correlation coefficient and the p -value in italic. Asterisks refer to significativity level with *: $p < 0,05$; **: $p < 0,01$; *** $p < 0,001$.

A

C. islandica	VOA	Growth	Photoperiod	Temperature
VOA		0.584 0.129	0.248 0.554	0.240 0.605
Growth			0.698 0.0544	0.672 0.0981
Photoperiod				0.781* 0.0381

B

Mytilus sp.	VOA	Growth	Photoperiod	Temperature
VOA		0.973*** 0.000046	0.918** 0.00130	0.710 0.0740
Growth			0.941*** 0.000501	0.775* 0.0408

Photoperiod				0.781* 0.0381
-------------	--	--	--	------------------

How was temporal autocorrelation accounted for in the linear models?

For the significant linear models (Fig. 2), we tested the residual temporal autocorrelation with Durbin-Watson test (J. Durbin & G.S. Watson (1950), Testing for Serial Correlation in Least Squares Regression I. *Biometrika* 37, 409–428. ; J. Durbin & G.S. Watson (1951), Testing for Serial Correlation in Least Squares Regression II. *Biometrika* 38, 159–178.).

The tests show (see below) that the coefficients of autocorrelation are not significant.

For information:

Mussels:

Growth vs VOA: DW = 1.8969, p-value = 0.4385

Growth vs photoperiod: DW = 2.8508, p-value = 0.910

Growth vs temperature: DW = 2.114, p-value = 0.5636

Scallops:

Growth vs photoperiod: DW = 1.6709, p-value = 0.3102

Were temperature and photoperiod tested in separate linear models, or were they both included in a single model?

We previously tested temperature and photoperiod included in a single model (Multiple linear regression).

For the scallop, the model was not significant ($R^2 = 0.55$, $p = 0.203$) and the p-value of each parameter, not significant as well ($p = 0.403$ for the photoperiod and 0.629 for the temperature).

For the mussel, the model was significant ($R^2 = 0.895$, $p = 0.011$). P-value was significant for the photoperiod ($p = 0.029$) and not significant for the temperature (0.729).

Consequently (also taken into account the Pearson tests), we decided to show separate linear models to highlighted the effect of the photoperiod (the correlation coefficient is higher and more significant for the photoperiod in separate linear model than in the multiple linear model)

The data in figure 2 D are suggestive of a non-linear relationship, and a polynomial predictor should be used to investigate the relationship for both species. This can be done using R.

To take into account the remark of the referee we did nonlinear regressions for the relation between the temperature and the growth for both species.

We did two kinds of nonlinear regressions. First, we applied a polynomial model (model 1: $f = y_0 + a * x + b * x^2$) and second an Exponential Rise to Maximum model (model 2: $f = a * (1 - \exp(-b * x))$)

For both models, no significant non-linear relationships were found. See below.

Scallop:

Model 1: $R^2 = 0.396$, $p = 0.0771$; Model 2: $R^2 = 0.288$, $p = 0.225$

Mussel:

Model 1: $R^2 = 0.449$, $p = 0.0546$; Model 2: $R^2 = 0.588$, $p = 0.0755$

Line 134 states that results for temperature and photoperiod were contrasting - contrasting what? The results complement each other for blue mussels; they both indicate positive correlations with mean daily shell growth (Fig. 2D, E). For the scallops, the results seem similar to blue mussels (Fig. 2E), especially if the relationship is non-linear.

The aim of this sentence and especially the term ‘contrasting’ was to say the relations between the growth and the VOA, the photoperiod and temperature were different between both species. We are not agreeing with the referee to say that for the scallops the results are similar to those of mussels. No nonlinear relationships were found. The statistics show that VOA is not correlated to the growth with scallops, on contrary for the mussels. The photoperiod effect on the growth is stronger for the mussels in comparison to the scallops. And finally, the temperature effect on the growth is weak for the mussels and non-significant for the scallops. See ln 137-147.

Please report R-square values and effect sizes for all linear models because p values do not provide information about the biological significance of a result.

The r-square and p-value of the linear models are already shown in the figure 2 (C-E). In the graphs, it is written in orange for *C. islandica* and in green for *M. edulis*. The r-square is also given with the p-value in the text (Results section) ln144-145, except when the p-value was not significant ($p=0.129$ and $p= 0.098$, ln145).

Now, we added as well the r-square, ln144-145.

Concerning the effective size used for the linear regression, it is added now in the legend of the figure 2 C-E. Ln 319.

Moreover, ln 147-150, p-values are done for the chronobiological analysis using the Cosinor model, but this model has not r-square values.

Reviewer: 2

Comments to the Author(s)

Review Tran et al.

This is a nice piece of work with a strait forward experimental set up. The only point which I do not understand in this set up is why the authors chose the scallop *Chlamys islandica* as examples for a native species and not a native mussel with a similar valve activity behavior to *Mytilus* sp.. This would have increased the informative value to what extent *Mytilus* is able to establish itself in higher latitudes. The choice of the organisms used should be explained in the text.

We thank the referee to her / his valuable comments.

Concerning the question about the comparison between a native species with the mussel *Mytilus*.

In Svalbard, where we undertook this study, the genus *Mytilus* is the only known mussel species. We understand the referee concern about the opportunity to compare other native mussels and *Mytilus*, but it was not possible. In Svalbard other bivalve species are present, such as *Arctica islandica* however *Arctica islandica* is a non-sessile clam, living in soft sediment / substrate therefore not occupying the same ecological niche as *Mytilus spp.* Other mussel species are present in the Arctic ocean around the polar circle, such as *Modiolus modiolus*. However, this specie is not present, at the moment, in high arctic locations such as Svalbard (This species was present in the past, around 8000 BP, but not currently). *C. islandica* is currently the only specie present in Svalbard that occupies a similar ecological niche as *Mytilus*, which is why this species was chosen to compare valve behavior. To justify the comparison of *Mytilus sp.* with *Chlamys islandica*, we modified the sentence In 64-71

Abstract

Row 37: ...in the high Arctic,...

Ok, we modified, ln 37.

Introduction

Row 52: ...in the Arctic...

Ok, we modified, ln 52.

Row 57...of Arctic ecosystem processes

Ok, we removed “the”, ln 57.

Results

Figure 1: It would be informative when on the right and left side are the month of the year

Ok, we modified the figure 1 accordingly.

Row 135:.....revealed contrasting results between the species.

Ok, we modified the sentence, ln 142-144.

Row 137: ...lesser to temperature.

Ok, we removed “the”, ln 144.

Row 139: ...with photoperiod.

Ok, we removed “the”, ln 148.

Discussion

Row 145:...show contrasting results..

Ok, we modified, ln 154.

Row 149:...we demonstrated. (to choose another word than always showed)

Ok, we modified, ln 158.

Row 151-156: This sentence is in combination with the following confusing. I would

recommend to remove the first sentence and start with: In *Mytilus* sp., we observed a clear annual pattern of shell growth that follows.....In the Arctic bivalve *Arctica islandica*, it was suggested that photoperiod is a cue for the annual valve behavior but weaker than food supply. This annual valve rhythm is shared...

Ok, accordingly to the referee, we modified the sentences, ln 153-157.

Conclusion

Row 202.204: “The effects of extreme photoperiods on the annual rhythm of this species is a negligible factor on the species northward expansion and reflects the species high capacity to deal with drastic light regimes”

We don't understand what the referee wants. This sentence above is exactly the same in the manus. However, we have slightly modified the sentence to be more comprehensive. Ln 209-211.

To conclude that *Mytilus* sp. Has a high capacity to deal with the drastic light regime is in my opinion only possible when comparing the data with native mussels of a similar valve behavior pattern. Therefore, the authors has to be clear that this is not a direct comparison between the two species (see comment in the intro. above), so that they should consider this point in their conclusion.

See comments above.

We cannot compare with other mussel species in Svalbard, but we feel that comparing its behavior with the scallop *C. islandica*, is relevant due to the characteristic shared in terms of food web position and fixation to the substrate.

We modified the sentence ln 64-71.